# Effect of balance training on footwork performance in badminton: An interventional study

**Kavinda T. Malwanage**[1]*, **Vindya V. Senadheera**[1], **Tharaka L. Dassanayake**[2,3]

**1** Department of Physiotherapy, Faculty of Allied Health Sciences, University of Peradeniya, Peradeniya, Sri Lanka, **2** Department of Physiology, Faculty of Medicine, University of Peradeniya, Peradeniya, Sri Lanka, **3** School of Psychological Sciences, The University of Newcastle, Callaghan, NSW, Australia

* ktmalwanage@pdn.ac.lk

## Abstract

Badminton is a racket sport that requires a wide variety of proficient postural changes and moves including jumps, lunges, quick changes in direction, and rapid arm movements. Efficient movement in badminton court entails reaching the shuttlecock in as few steps as possible while maintaining good balance. Balance training is an unexplored component in badminton training protocol, though balance is important in injury prevention and performance enhancement. We aimed to investigate the effectiveness of balance training on sport-specific footwork performance of school-level competitive badminton players. We conducted a controlled trial involving 20 male badminton players (age 12.85±0.67 years). Participants were stratified according to their level of performance in the game, and payers from each stratum were randomly assigned to control and intervention groups. The control group (n = 8) engaged in 2 hours of ordinary badminton training, whereas the intervention group (n = 12) underwent 30 minutes of balance training followed by 1 hour and 30 minutes of ordinary badminton training, 2 days per week for 8 weeks. We tested the participants at baseline and after 8 weeks for static balance (Unipedal Stance Test), dynamic balance (Star Excursion Balance Test) and sport-specific footwork performance (shuttle run time and push-off times during stroke-play). On pre- vs. post-intervention comparisons, both groups improved in static balance (eyes opened) (p<0.05), but only the intervention group improved in dynamic balance (p = 0.036) and shuttle-run time (p = 0.020). The intervention group also improved push-off times for front forehand (p = 0.045), side forehand (p = 0.029) and rear around-the-head shots (p = 0.041). These improvements in push-off times varied between 19–36% of the baseline. None of the footwork performance measures significantly improved in the control group. Our findings indicate that incorporating a 30-minute balance training program into a regular training schedule improves dynamic balance, and on-court sport-specific footwork performance in adolescent competitive badminton players, after 8 weeks of training.

**Data Availability Statement:** All relevant data are within the paper and its Supporting Information files.

**Funding:** The author(s) received no specific funding for this work.

**Competing interests:** The authors have declared that no competing interests exist.

## Introduction

Badminton is a popular sport where participants of all ages can engage at competitive level. Aerobic stamina, agility, strength, explosive power, speed, flexibility, balance and coordination are the main components of physical fitness required by a badminton player in order to be proficient in the sport [1, 2]. The game is characterized by high racket and shuttlecock speed, with the shuttle struck at over 250 km/h at elite level encounters. Decreased effective playing time and increased shot frequency in Olympic level badminton matches recorded over the last few decades signify that the game has increased in speed with time [3–6]. To succeed in the game under these increasing demands, the players require extremely fast reaction speed, agility and quickness to display the highest levels of athleticism [1, 2].

Competitive badminton engages a wide variety of postural changes including jumps, lunges, quick changes in direction, and rapid arm movements. Movement in the badminton court requires reaching the shuttle in as few steps as possible while maintaining good balance and keeping the body under control [1, 2]. Stroke play and footwork performance have been identified as two fundamental skills in badminton where the stroke play is influenced by eye-hand coordination and the footwork is influenced mainly by balance [1, 2].

Footwork performance is characterized by the ability to accelerate or decelerate and change directions on the court for accurate shots and better performance [1, 2]. Footwork includes moving to and from six zones of the court (viz. right and left frontcourt, the right and left mid-court, and the right and left rear court) [7] using different stepping strategies, lunge strategies and arm movements. Footwork agility is pertinent in badminton performance [8] hence it is always emphasized during training and playing.

Footwork training in badminton includes shuttle run and shadow play [1, 2]. Assessment of footwork performance is crucial to evaluate badminton performance. To date, assessment of footwork performance is mainly done by kinetic and kinematic analysis of lunge performance such as velocity at touchdown [9], vertical ground reaction force [9–14] peak pressure, maximum force and contact area of the foot [15], knee and ankle range of motion [10, 11], foot strike angle [14], and plantar pressure analysis [15–17]. Spatiotemporal parameters such as approaching speed [12–14, 17], total duration and recovery duration [12], foot contact time [13, 14], and heel landing time [16] have also been used to assess the badminton lunge performance. Quick recovery from a stroke is an important aspect in badminton footwork; and push-off time in a stroke (i.e., the time from the point of initial contact to clear off the foot from the ground during an effective shot) is identified as a main determinant of such quick recovery [1, 2].

Lunges account for approximately 15% of all movements in the badminton court [12]. The players should competently perform lunges in longitudinal, diagonal and transversal directions during a match [18]. Lam et al. (2020) reported that elite badminton players demonstrate good lunge performance with more aggressive knee and ankle strategy, and this correlates with their chances of winning the game [19]. Kinematic and kinetic analysis revealed that badminton lunges subject the hip and ankle joints to high torques. Maintaining a good balance and a posture counteract these forces during a lunge and prepare the player for the next shot [20]. Lunges place high physical demands on the lower limbs, and badminton players were found to have calcaneal bone stiffness and higher mean plantar pressure than healthy controls [21]. However, the adaptations seem to be asymmetrical: professional badminton players have thicker patellar and Achilles tendons in their dominant leg compared to the non-dominant leg [22]. In addition to lunges, rear court movements in badminton also require the player to have highly developed balance. Plantar pressure analysis shows that, badminton players usually are in contact with the ground over the forefoot without the midfoot and heel during rear court

forehand strokes [23]. Irrespective of the strokes they offer, the players need to maintain their center of gravity within the base of support, not to lose balance and to move in any direction after returning the shuttlecock. Elite badminton players are observed to possess minimal fluctuations in center of gravity compared to amateur badminton players [24].

These findings suggest that balance is a vital aspect of the footwork performance to move across the court faster. The ability to maintain dynamic balance has been directly linked to better control of jumping and running to smash, and making the lunges [25]. Apart from sport-specific role of balance in badminton, generic importance of balance training in injury prevention across different sports have also been well established [26–28]. The overall evidence thus indicates that improvements in static and dynamic balance could 1) shorten the recovery time from a given stroke (defined as push-off time in badminton) and 2) enable the player to move across the court to the next shot (assessed on-court using shuttle run).

Balance training has been found to enhance joint sequential action chain of upper limbs in badminton stroke play [29]. Even though balance is identified as an important concept and footwork performance is identified as a fundamental skill in high performance in badminton [1, 2], to our knowledge no study has examined the effect of balance training on footwork performance in badminton. This study aimed to fill that hiatus in evidence by investigating the effectiveness of balance training on on-court footwork performance of badminton players. Specifically, in the present study, we investigated the effect of an 8-week balance training program on generic measures of static balance (viz. Unipedal Stance Test) and dynamic balance (viz. Star Excursion Balance Test); and badminton-specific measures of on-court footwork performance (viz. shuttle run time and push-off times).

## Materials and methods

### Study design and setting

This was a controlled trial that was conducted in an indoor badminton stadium of a school in the Kandy District in Sri Lanka from January 2018 to April 2018. Static and dynamic balance tests and shuttle run tests were performed in each participant. Further, on-court footwork performance for seven badminton shots were analyzed.

### Participants

Competitive male badminton players in the age group of 13–15 years were screened for recruitment for the study. Thirty healthy players who were engaged in regular badminton training for more than 6 months were recruited initially as participants. Players with a history of acute or chronic musculoskeletal disorders were excluded. Participant enrolment, allocation, intervention and follow up procedure (CONSORT flow diagram) is presented in Fig 1. Participants were stratified based on their level of performance in their age group into national level, provincial level, zonal level and school level players. Then the participants in each stratum were randomly assigned to either the *intervention group (n = 15)* or the *control group (n = 15)*, by applying simple random sampling for each stratum by the investigators. Ethical clearance for the study was obtained from the Ethics Review Committee of Faculty of Allied Health Sciences of University of Peradeniya (Ethical approval number AHS/ERC/2018/005). Participants were informed of the benefits and risks of the investigation prior to the recruitment and the written informed consent was obtained from all participants and their parents. The collected data were kept confidential, to which only the investigators had access. Except consent documents, all other data documentation was anonymized and had only the subject numbers.

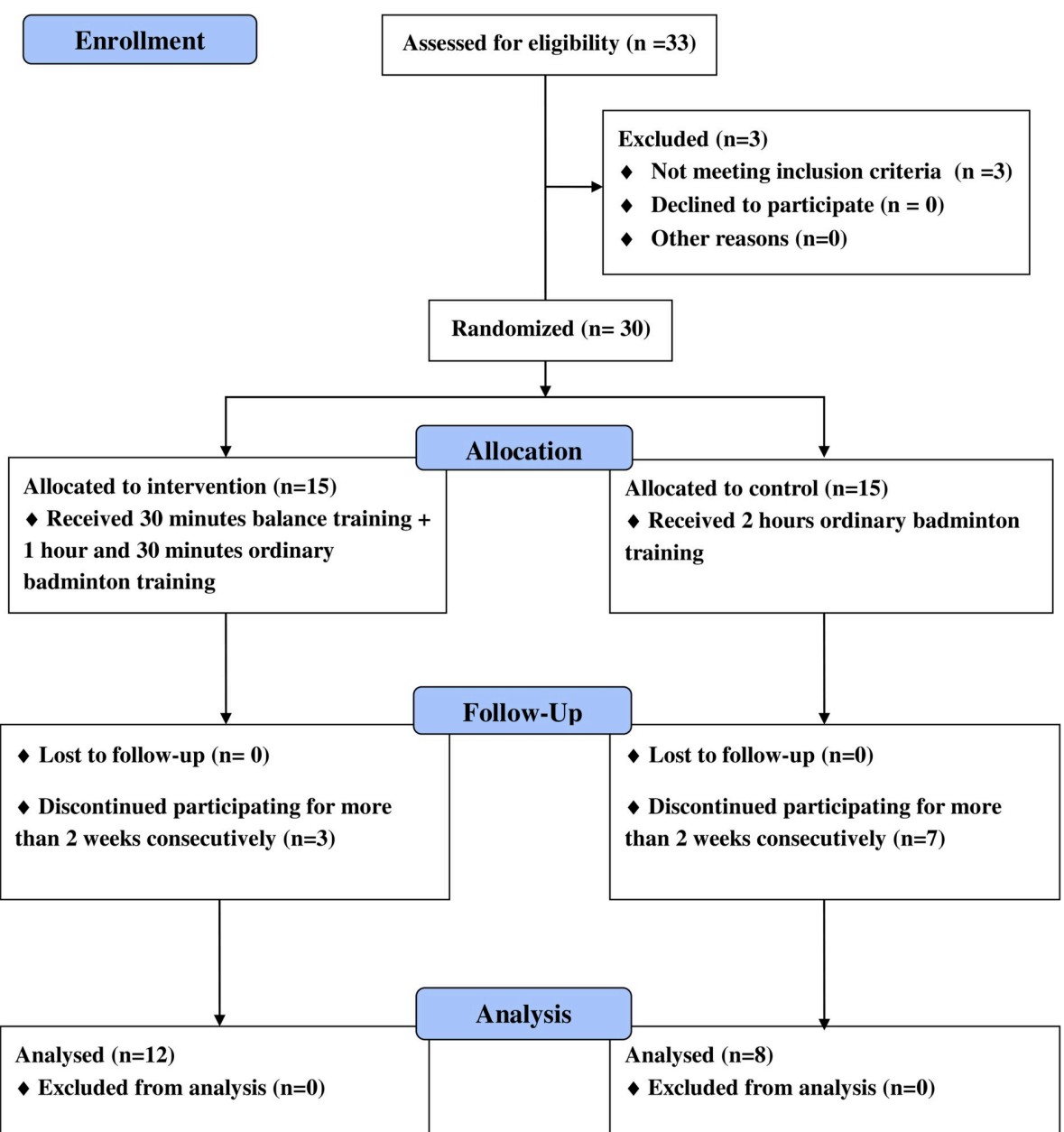

**Fig 1. CONSORT flow diagram of participant's selection.** When randomly assigning the participants from strata, two participants were randomly assigned into the control group and three into the intervention group under "Provincial level" stratum, whereas seven participants were randomly assigned into the control group and six into intervention group respectively under "School level" stratum. Equal numbers of participants were randomized to each group in the other two strata.

## Study protocol and intervention

The training protocol undergone by each group of participants during each session is illustrated in Fig 2. The control group engaged in a 2-hour ordinary badminton training, 2 days per week for 8 weeks, conducted by their coach who had Badminton World Federation (BWF) coach Level 2 certification. The intervention group was engaged in a specially designed 30-minute balance training program conducted by the researchers, before a 90-minute ordinary badminton training session conducted by the same coach, 2 days per week for 8 weeks.

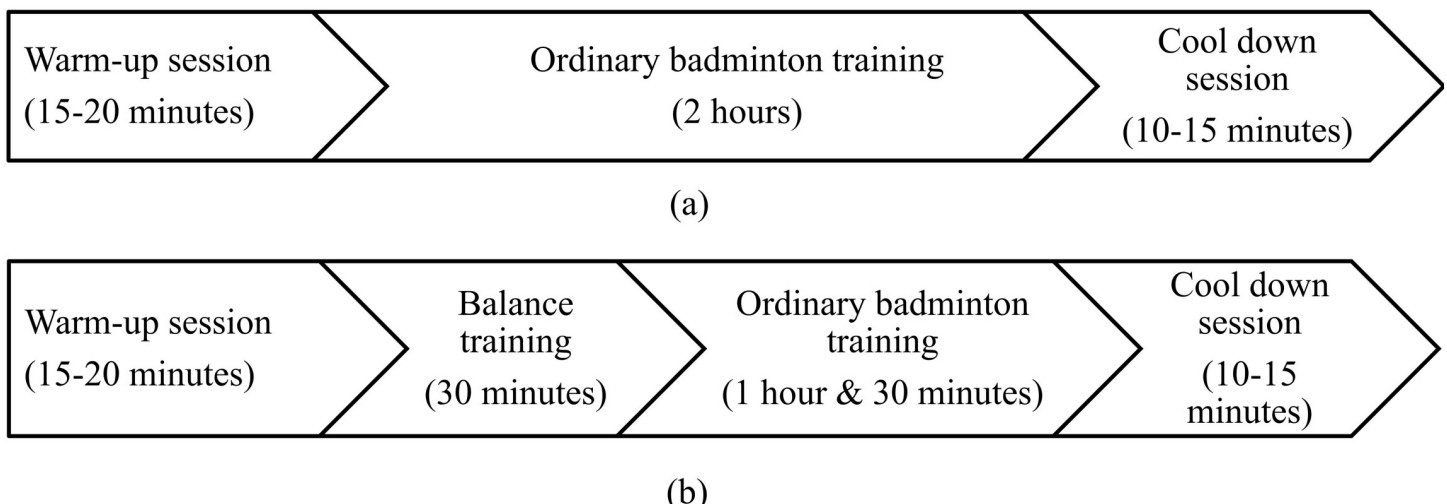

**Fig 2.** Flow of the training protocol in each day: (a) control group (b) intervention group.

After the 30 minutes of balance training, the intervention group started off the ordinary badminton training as if they are starting a new badminton session on a regular training day. As such, the 30 minutes they lost was from the latter end of the day's session, where the control group players used to play practice matches etc. with their playmates.

The 8-week balance training program was developed and conducted by two investigators who are registered physiotherapy professionals (KTM and VVS); and it consisted of a set of evidence-based exercises [30, 31] aimed to improve static and dynamic balance through different genres of balance exercises (S1 File). Owing to the fact that balance is a complex motor control task involving visual, proprioceptive and vestibular systems, this balance training program was focused on all those three aspects of the intervention group.

Participants of intervention group engaged in balance exercises which required (1) to assume different stances including single leg, double leg, and tandem leg; (2) different lower limb activities including lunging in different directions, marching and walking; and (3) different arm positions to use support surface, hands on waist or folded across chest. The difficulty of each exercise was achieved by changing the type of surface on which the participants underwent the exercises from the floor to the Both Sides Up (BOSU) balance trainer to the wobble board. In the last set of exercises, participants were asked to simultaneously engage in upper limb tasks related to badminton while maintaining balance on the BOSU balance trainer. The balance training protocol was devised to increase the difficulty in every 2 weeks during the 8-week program. In each 2-week period, the participants underwent 4–5 different balance exercises including at least one exercise from each of the above categories. As the training progressed, the static and dynamic balance exercises were modified to make them more demanding by decreasing base of support, increasing center of gravity sway on base of support, removing visual feedback (eyes closed) and changing the type of the surface on which they stand on and perform the exercises (for instance, getting them to do static balance exercises and lunges initially on the floor, and them moving onto the BOSU balance trainer and then to the wobble board). The detailed biweekly protocol is shown in S1 File.

### Tests and measurements

Weight and height of each participant were measured at the recruitment. Each participant's dominant leg was identified as the leg which moved close to the racket arm when lunging. It

was found that right limb was dominant in all the participants except one. Leg length of dominant lower limb was measured in centimeters using a standard measuring tape, from anterior superior iliac spine to the medial malleolus of tibia with the participant in the supine position. The outcome measures were performance of two standard tests of static balance (Unipedal Stance Test) and dynamic balance (Star Excursion Balance Test) and two measures of on-court footwork performance (i.e., shadow play) viz. shuttle run time and push-off time during a stroke -play routine. These outcome measures were measured before the balance training program (baseline) and at the end of 8 weeks of training.

**Unipedal stance test.** Unipedal Stance Test [32] was used to assess static balance of each participant by one investigator who is a qualified physiotherapist. During the test, participants were asked to stand barefoot on the limb of their choice, with the other limb raised so that the raised foot positioned near but not touching the ankle of their stance limb, with arms crossed over the chest. Each participant was instructed to focus on a spot on the wall at eye level in front of him, for the duration of the eyes open test. Time calculation commenced using a stopwatch when the participant raised the foot off the floor. Time calculation was ended when the participant either: (1) used his arms (i.e., uncrossed arms); (2) used the raised foot (i.e., moved toward or away from the standing limb or touched the floor); (3) moved the weight-bearing foot to maintain balance (i.e., rotated foot on the ground); (4) a maximum of 45 seconds had elapsed, or (5) opened eyes on eyes-closed trials. The procedure was repeated 3 times and the best of the three trials was recorded. Participants performed 3 trials with the eyes opened and 3 trials with the eyes closed, alternating between the conditions. A 5-minute rest was allowed between each trial set to avoid fatigue.

**Star excursion balance test.** Star Excursion Balance Test [33] was carried out to assess the dynamic balance of each participant by one investigator, who is a qualified physiotherapist. During the test, the participants were asked to stand on the non-dominant leg in the center of the grid of 8 lines with hands on their iliac crests. Then participants were asked to reach in the clockwise as far as possible along the 8 reaching directions: anterior; anterior-lateral; lateral; posterior-lateral; posterior; posterior-medial; medial; anterior-medial. Participants were instructed to lightly touch the line with the most distal part of the reaching foot, and return the reaching leg back to double-leg stance, while maintaining a single-leg stance with the other leg in the center of the grid. The reach distances were recorded with a mark on the tape line at the point of maximal reach and were measured from the center of the grid. The average of three trials was normalized by dividing by the previously measured leg length to standardize the maximum reach distance. A trial was discarded and repeated if the investigator noted the participant using the reaching leg for a substantial amount of support at any time, removed the weight-bearing foot from the center of the grid, or was unable to maintain balance on the support leg throughout the trial [34].

**Shuttle run.** The shuttle run is a standard coaching phenomenon and a practice-based protocol [1, 2]. Specifically, the shuttle run mimics sport-specific footwork and therefore best considered a proxy-measure of badminton footwork performance. In the shuttle run circuit we set up in the present study, three shuttlecocks were initially placed in the forehand corner of the forecourt. Participant started picking up each shuttle from forehand forecourt and placed in the backhand forecourt corner, then the forehand sideline, then the backhand sideline, then the forehand backcourt followed by backhand backcourt corner. Each movement was made via the base position at the center-court using the same footwork routine used in real-full court play thus mimicking the footwork in a real game. Total time taken to accomplish all the movements in six directions was recorded. The average of the three shuttle runs was taken as the shuttle run time. A 6-minute rest was given between two shuttle runs.

**Measurement of push-off time in stroke play.**   To evaluate the push-off times, twelve shuttlecocks were fed randomly to six primary locations of the court by the coach (two shuttles per each location). The participants offered seven different shots to the shuttles fed to them during the rally: front forehand, front backhand, side forehand, side backhand, rear forehand, rear backhand and rear around the head. It should be noted that the coach feeding the shuttles led to an unavoidable human error, as to the exact position of the court where the shuttle was intercepted by the participant. However, we believe that this only increased the random error of the point of intercept of the shuttle (and potentially the push-off time) across participants and sessions, without causing a systematic bias. The routine, of each participant was recorded using GoPro Hero 5 high speed video camera (Woodman labs, Inc, CA, USA, 2016). The video was acquired at a rate of 120 frames per second with a screen resolution of 720 × 1080 pixels with 12 effective megapixels as an MP4 file. Using this video, the push-off time of each shot was analyzed using Kinovea 0.8.15 motion analysis software (Joan Charmant & Contributors, Belgium, 2006), and recorded in milliseconds. Push-off time was measured as the time from the point of initial contact to clear off the foot from the ground during an effective shot. To be qualified as an effective shot, the participant should have contacted the shuttlecock with the badminton racket.

**Statistical analysis.**   Shuttle run time, push-off times for different shots and measures of static balance with eyes opened, static balance with eyes closed and dynamic balance were considered the outcome measures of interest. All participants completed all the balance tests. However static balance with eyes closed data of one participant from the intervention group and one participant from the control group had to be excluded because of being outliers. Shuttle run data were based on all participants. Footwork performance data were incomplete for some shots and sessions because the participants either (1) failed to perform desired badminton shot, or (2) unable to contact the shuttle. Consequently, the numbers completed under each stroke were different. Particularly, this caused a major reduction in samples for front backhand, side backhand and rear forehand shots. These data are also presented with the rest, but we did not make any interpretations based on those limited data.

Two-way, group (intervention vs. control) × time (pre vs. post) mixed analysis of variance (ANOVA) was conducted on all the outcome measures to evaluate the effect of intervention compared to the regular training routine. The level of significance was ascertained at a cut off p value of less than 0.05. All statistical analyses were performed using IBM SPSS 22.0 software (Armonk, NY: IBM Corp).

## Results

### Sample characteristics

Of the sample of 30 participants, there were seven dropouts from the control group and three dropouts from the intervention group at the end of 8 weeks of the trial. Eight participants in the control group and 12 participants in the intervention group completed the training program. Accordingly, data from 8 participants in control group and 12 participants in intervention group were analyzed. The final sample (n = 20) had a mean age of 13.85 (SD = 0.67) years, mean height of 1.48 (SD = 0.09) m, mean weight of 36.77 (SD = 10.09) kg, and a mean Body Mass Index (BMI) of 16.61 (SD = 3.04). Table 1 presents the characteristics of the control and the intervention groups.

### Balance performance and shuttle-run time

The time × group ANOVA and paired pre-post intervention comparisons in balance tests and the shuttle-run time are shown in Table 2. Static balance performance with eyes opened showed a time main effect (p = 0.001) with both the intervention group (p = 0.013) and the

**Table 1. Sample characteristics.**

| Measure | Control group (n = 8) | Intervention group (n = 12) | Mean difference [95% CI] | *p* value |
|---|---|---|---|---|
| Age (years) (mean ± SD) | 13.00 ± 0.93 | 12.75 ± 0.45 | 0.25 [-0.40, 0.90] | 0.593 |
| Height (m) (mean ± SD) | 1.49 ± 0.11 | 1.47 ± 0.09 | 0.02 [-0.08, 0.11] | 0.244 |
| Weight (kg) (mean ± SD) | 36.48 ± 9.48 | 36.97 ± 10.89 | -0.49 [-10.43, 9.45] | 0.440 |
| BMI (mean ± SD) | 16.29 ± 2.50 | 16.82 ± 3.44 | -0.53 [-3.51, 2.45] | 0.224 |

control group (p = 0.023) showing an improvement of around 7 seconds. However, when the eyes were closed static balance did not improve significantly in either group. Dynamic balance showed a significant time main effect (p = 0.009), but only the intervention group showed a significant improvement (p = 0.036). Pre-post comparisons of shuttle run time showed a 2.65-second improvement (p = 0.020) in the intervention group, but no improvement in the control group (p = 0.918).

## Footwork performance

The pre-post intervention comparisons of push-off times for the seven badminton shots are summarized in Table 3 and Fig 3. Mean push-off times in all badminton shots were reduced in the intervention group after the balance training program, although only three were statistically significant. These included front forehand (p = 0.045), side forehand (p = 0.029) and rear around-the-head (p = 0.041) shots, amounting to a reduction by 35.79%, 29.76% and 18.89%, respectively of the pre-intervention push-off times. In contrast, push-off times of none of the shots improved in the control group (Table 3). The data for front backhand and rear forehand badminton shots were not considered for further interpretations only a few participants could perform those shots.

## Discussion

In this study, we investigated the effectiveness of a balance training program on improving static and dynamic balance, and sport-specific footwork performance of adolescent

**Table 2. ANOVA and paired comparisons in balance tests and the shuttle-run time.**

| Outcome measure | Two-way ANOVA significance (p values) | | | Within-group pre-post comparisons | | | | | | | | | |
| | | | | Control group | | | | | Intervention group | | | | |
| | Group | Time | Time × Group | n | Pre-test mean ± SD | Post-test mean ± SD | Post-Pre difference [95% CI] | *p* value | n | Pre-test mean ± SD | Post-test mean ± SD | Post-Pre difference [95% CI] | *p* value |
|---|---|---|---|---|---|---|---|---|---|---|---|---|---|
| Static balance with eyes opened | 0.520 | 0.001 | 0.918 | 8 | 7.35 ± 4.25 | 14.59 ± 5.34 | 7.24 [1.31, 13.17] | 0.023* | 12 | 8.48 ± 5.87 | 15.36 ± 4.18 | 6.88 [1.76, 11.99] | 0.013* |
| Static balance with eyes closed | 0.240 | 0.110 | 0.352 | 7 | 2.72 ± 2.04 | 4.56 ± 1.19 | 1.84 [-0.37, 4.04] | 0.089 | 11 | 3.27 ± 1.86 | 9.93 ± 13.65 | 6.66 [-2.23, 15.56] | 0.128 |
| Dynamic balance | 0.379 | 0.009 | 0.802 | 8 | 79.52 ± 8.40 | 85.73 ± 11.85 | 6.21 [-1.52, 13.93] | 0.099 | 12 | 82.78 ± 9.49 | 90.17 ± 12.11 | 7.39 [0.58, 14.2] | 0.036* |
| Shuttle-run time | 0.460 | 0.114 | 0.085 | 8 | 49.26 ± 7.04 | 49.39 ± 5.57 | 0.13 [-2.58, 2.82] | 0.918 | 12 | 48.84 ± 5.48 | 46.19 ± 4.22 | - 2.65 [-4.79, -0.50] | 0.020* |

Static balance with eyes opened, static balance with eyes closed and shuttle run time are measured in seconds. Dynamic balance represents the normalized reach distance which is expressed as a percentage of leg length.

*$p < 0.05$, compared with control group.

**Table 3. ANOVA and paired comparisons of push-off times (in milliseconds) for seven badminton shots.**

| Badminton shot | Two-way ANOVA significance (p values) | | | Within-group pre-post comparisons | | | | | | | | | |
| --- | --- | --- | --- | --- | --- | --- | --- | --- | --- | --- | --- | --- | --- |
| | | | | Control group | | | | | Intervention group | | | | |
| | Group | Time | Time× Group | n | Pre-test mean ± SD (ms) | Post-test mean ± SD (ms) | Post-Pre difference [95% CI] (ms) | p value | n | Pre-test mean ± SD (ms) | Post-test mean ± SD (ms) | Post-Pre difference [95% CI](ms) | p value |
| Front forehand | 0.024 | 0.121 | 0.160 | 8 | 258.56 ± 68.83 | 254.13 ± 74.69 | - 4.44 [-96.50, 87.63] | 0.912 | 11 | 236.31 ± 112.43 | 151.73 ± 41.84 | - 84.58 [-166.66, -2.50] | 0.045* |
| Front backhand | 0.918 | 0.864 | 0.386 | 2 | 187.25± 18.03 | 329.0 ± 171.12 | 141.75 [-1557.71, 1841.21] | 0.482 | 1 | 342 | 150 | -192 - | - |
| Side forehand | 0.109 | 0.494 | 0.019 | 8 | 205.48 ± 59.38 | 244 ± 73.65 | 38.53 [-34.12, 111.17] | 0.250 | 12 | 226.38 ± 78.03 | 159 ± 24.87 | - 67.38 [-126.41, -8.34] | 0.029* |
| Side backhand | 0.306 | 0.914 | 0.930 | 4 | 234.38 ± 280.66 | 168.0 ± 83.84 | - 66.38 [-574.57, 441.82] | 0.706 | 8 | 246.75 ± 138.17 | 168.25 ± 57.13 | - 78.5 [-210.78, 53.78] | 0.203 |
| Rear forehand | 0.618 | 0.278 | 0.460 | 1 | 275.0 | 292.0 | 17 - | - | 2 | 183.0 ± 82.02 | 114.50 ± 20.51 | - 68.5 [-621.22, 484.22] | 0.360 |
| Rear backhand | 0.908 | 0.179 | 0.518 | 8 | 70.75 ± 131.32 | 40.63 ± 114.91 | - 30.13 [-190.97, 130.72] | 0.671 | 12 | 93.38 ± 159.00 | 9.67 ± 33.49 | - 83.71 [-190.73, 23.31] | 0.113 |
| Rear around-the-head | 0.133 | 0.850 | 0.064 | 8 | 162.29 ± 70.15 | 196.63 ± 50.34 | 34.34 [-55.78, 124.45] | 0.398 | 11 | 220.8 ± 51.21 | 179.09 ± 25.07 | - 41.71 [-81.46, -1.96] | 0.041* |

*$p <0.05$, compared with control group.

competitive badminton players. We observed significant improvements in static balance with eyes opened in both groups. Only the intervention group showed significant improvements in dynamic balance, shuttle run time and push-off times of front and side forehand shots, and rear around-the-head shots. Push-off times for none of the seven shots improved in the control group. Research on the impact loading characteristics of badminton lunge reveals that the unskilled players exhibit longer contact time and smaller foot strike angle compared to skilled players [14]. Given that proficient competitive athletes demonstrate greater balance ability than their less proficient counterparts [35], our findings indicate balance training is a useful adjunct to regular training schedules of non-elite competitive athletes in enhancing their sport-specific performance. There is evidence that balance training improves agility run time in recreationally active young people [36], indicating the benefits may not be limited to competitive athletes.

It is recommended that sport-specific skill training should be introduced to boys around the age of 14 years which is the onset of 'peak height velocity'. This is the optimal stage of growth and development to introduce programs to enhance 'five Ss' of training and performance, viz. stamina, strength, speed, flexibility and technical skills [37]. Our findings,in a sample of 13-15-year-old competitive badminton players corroborate these recommendations. Complementing our findings, others have found that balance training also improves efficient proximal-distal joint sequential action chain in upper limbs in badminton [29]. In line with this, better footwork performance has also found to improve the upper limb performance by reducing the time spent to elevate the racket arm [38]. Such enhancement enables the player to intercept the shuttlecock early and launch the shuttlecock at a higher velocity.

Multiple studies implicate poor balance skills in sport injuries [25, 39–42]. Previous studies report that lower limb is the most susceptible area for badminton-related injuries [43–45], according to some studies, accounting to 40% of the injuries in badminton players [46]. It has been shown that balance training can be used prophylactically or after acute ankle sprain to

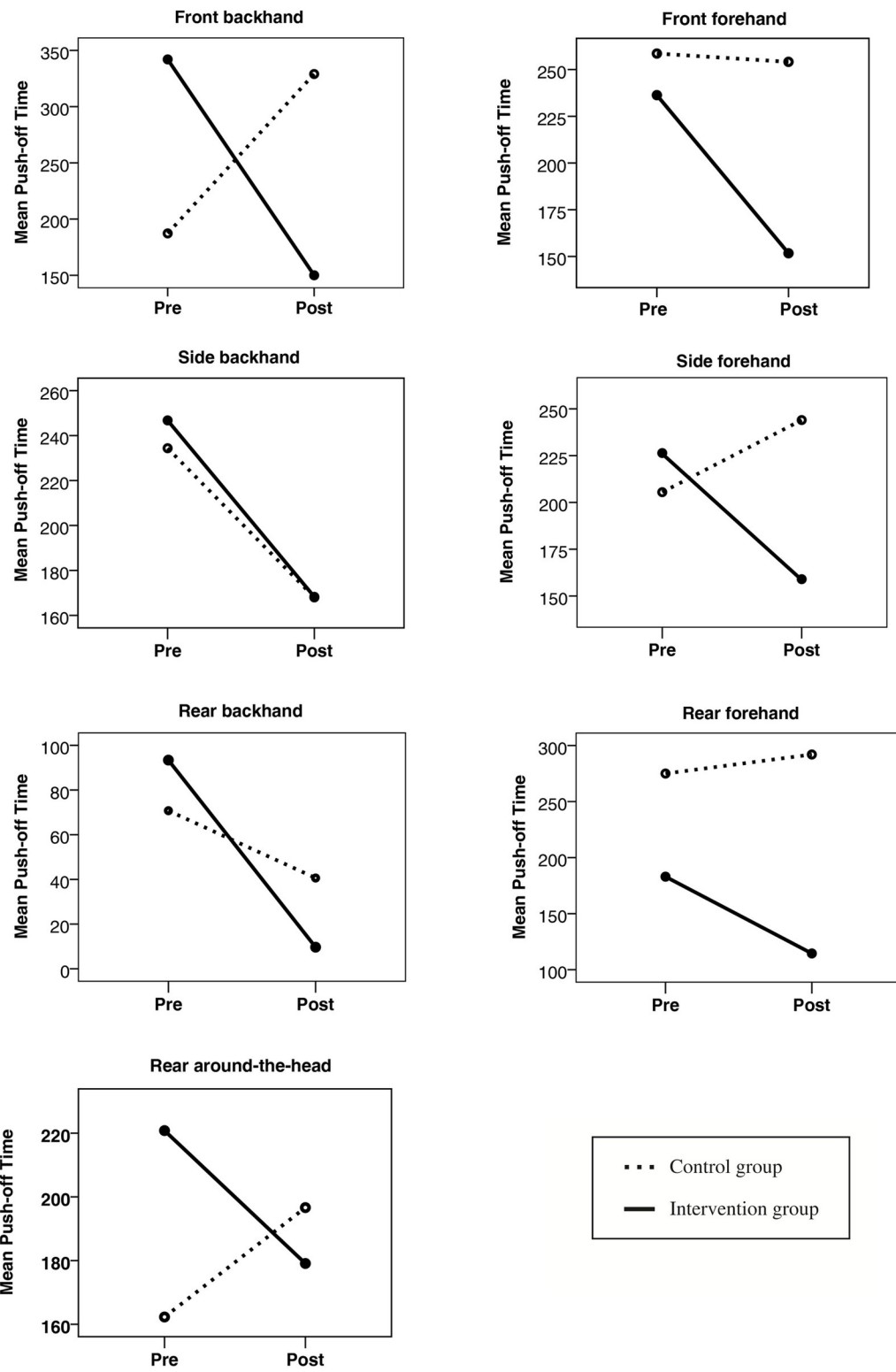

**Fig 3. Pre- and post-intervention mean push-off times (in milliseconds) for different badminton shots in the control group and the intervention group.** Pre = pre-intervention, Post = post-intervention.

reduce the risk of future ankle sprain [47]. Therefore, apart from enhancing sporting performance, balance training in badminton also improves the safety of the game. In line with our findings, a recent systematic review suggests that, to be effective in injury prevention and improving postural control in athletes, a balance training program should last at least for 8 weeks, with at least two sessions per week [27].

Some limitations of this study have to be acknowledged. The selection of the study sample was largely governed by the feasibility to conduct a uniform balance training program to a group of sportsmen in a closely supervised manner. From a scientific perspective we also wanted to control interrater variability/inter-coach variability which could occur as a result of different coaching styles, had we selected different training squads. This however limited the sample size and thus the power of the study. The limitation of the samples further restricted our ability to compare push-off time data for some strokes where there were considerable proportions of missing data. Still, we believe based on our results this training and assessment model can be implemented in larger, elite level cohorts on badminton. In that sense the present trial would also serve as a proof-of-concept study for the future.

## Conclusions

To the best of our knowledge, this is the first controlled trial to report the effects of balance training in enhancing footwork performance in adolescent competitive badminton players. Specifically, we observed replacement of 30 minutes of regular badminton training with a specially designed 30-minute balance training program improves push-off times in sport-specific footwork in adolescent competitive badminton players by 19%-36%, after 8 weeks of training.

Based on the training protocol that we administered, we believe that the balance training program should consist of different genres of static and dynamic balance exercises of different stances, lower limb activities and upper arm positions. As the training progresses, the balance exercises can be made more challenging by decreasing base of support, increasing center of gravity sway on base of support, removing visual feedback and changing the type of the surface on which the athletes stand on and perform the exercises. Thus, the coaches and physical trainers should pay their attention to incorporate a balance training program to improve the footwork performance of the badminton players. Future studies can also examine whether balance training enhances performance among beginners in middle childhood and adult competitive badminton players.

## Supporting information

**S1 File. Balance training protocol.**
(PDF)

## Acknowledgments

We thank Ms. D.M.S.Dissanayaka, the badminton coach of Sri Chandananda Buddhist College, Kandy, Sri Lanka for her assistance in implementing the balance training program and in data collection.

## Author Contributions

**Conceptualization:** Kavinda T. Malwanage, Vindya V. Senadheera.

**Data curation:** Kavinda T. Malwanage, Tharaka L. Dassanayake.

**Formal analysis:** Kavinda T. Malwanage, Tharaka L. Dassanayake.

**Investigation:** Kavinda T. Malwanage, Vindya V. Senadheera, Tharaka L. Dassanayake.

**Methodology:** Kavinda T. Malwanage, Vindya V. Senadheera, Tharaka L. Dassanayake.

**Project administration:** Kavinda T. Malwanage.

**Resources:** Kavinda T. Malwanage, Vindya V. Senadheera, Tharaka L. Dassanayake.

**Software:** Kavinda T. Malwanage, Vindya V. Senadheera, Tharaka L. Dassanayake.

**Supervision:** Tharaka L. Dassanayake.

**Validation:** Kavinda T. Malwanage, Vindya V. Senadheera, Tharaka L. Dassanayake.

**Visualization:** Kavinda T. Malwanage, Vindya V. Senadheera, Tharaka L. Dassanayake.

**Writing – original draft:** Kavinda T. Malwanage, Vindya V. Senadheera, Tharaka L. Dassanayake.

**Writing – review & editing:** Kavinda T. Malwanage, Vindya V. Senadheera, Tharaka L. Dassanayake.

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
