## [Decision Letter · Decision Letter 0]

22 Aug 2022

PONE-D-22-21470Effect of balance training on footwork performance in badminton: an interventional studyPLOS ONE

Dear Dr. Malwanage,

Thank you for submitting your manuscript to PLOS ONE. After careful consideration, we feel that it has merit but does not fully meet PLOS ONE’s publication criteria as it currently stands. Therefore, we invite you to submit a revised version of the manuscript that addresses the points raised during the review process.

ACADEMIC EDITOR: Dear Authors,

two experts in the field revised your current manuscript and recognised some points that should be addressed.

We look forward to receiving your revised manuscript.

Kind regards,

Javier Abián-Vicén, Ph.D.

Academic Editor

PLOS ONE

Journal Requirements:

Reviewers' comments:

Reviewer's Responses to Questions

**Comments to the Author**

1. Is the manuscript technically sound, and do the data support the conclusions?

Reviewer #1: Partly

Reviewer #2: Partly

2. Has the statistical analysis been performed appropriately and rigorously? 

Reviewer #1: Yes

Reviewer #2: I Don't Know

3. Have the authors made all data underlying the findings in their manuscript fully available?

Reviewer #1: Yes

Reviewer #2: Yes

4. Is the manuscript presented in an intelligible fashion and written in standard English?

Reviewer #1: Yes

Reviewer #2: Yes

5. Review Comments to the Author

Reviewer #1: The approach of the paper is interesting because it examines the effect of balance training on footwork performance in badminton but I think the authors should respond to the comments below in order to improve the quality of the paper. Despite I think the authors should increase the number of the sample (they have use only 8 in control group and 12 in the intervention group) I propose a major revision of the manuscript to be considered for publication (see the following comments).

- In the introduction section important references are missed. In the first paragraph when discussing the timing structure in badminton are missed 3 important references (i.e. Abian-Vicen et al., 2013; Abián, 2014; Abián-Vicén et al., 2018). In the second paragraph when are talking about that lunges place high physical demands on the lower limbs are missed 2 important references (ie. Bravo-Sánchez et al, 2019; Bravo Sánchez et al., 2019). These articles should be included.

Abian-Vicen J, Castanedo A, Abian P, & Sampedro J. (2013). Temporal and notational comparison of badminton matches between men's singles and women's singles. Int J of Perf Anal Sport, 13(2), 310-320.

Abián P, Castanedo A, Feng XQ, Sampedro J, & Abian-Vicen J. (2014). Notational comparison of men's singles badminton matches between Olympic Games in Beijing and London. Int J of Perf Anal Sport, 14(1), 42-53

Abián-Vicén J, Sánchez L, & Abián P. (2018). Performance structure analysis of the men’s and women’s badminton doubles matches in the Olympic Games from 2008 to 2016 during playoffs stage. International Journal of Performance Analysis in Sport, 1-12. doi: 10.1080/24748668.2018.1502975

Bravo-Sánchez, A., Abián, P, Jiménez, F, Abián-Vicén, J. (2019). Myotendinous asymmetries derived from the prolonged practice of badminton in profesional players. Plos One, 14(9): e0222190. doi: 10.1371/journal.pone.0222190

Bravo-Sánchez, A, Abián-Vicén, J, Jiménez, F, Abián, P. (2019). Influence of badminton practice on calcaneal bone stiffness and plantar pressure. Physician and Sportmedicine, 48(1), 98-104. doi: 10.1080/00913847.2019.1635050

- The badminton training should be explain to both group to know the difference exercises in the control group during the 30 minutes that is longer the session of badminton training compared to the intervention group.

-The shuttle run test and the testing for footwork performance have not been validated for any author.

- It is impossible to know that the shuttle is hit for all the participant in the same point (zone) during the testing for footwork performance. The shuttle is feeded by a person and in variables recorded in milliseconds the shuttle should be exactly in the same point for all the participants.

- In the results of the tests (in the tables) you can see that not all the participants have carried out all the tests.

- In table 3 in some variables there are 1 or 2 participants???. That is wrong.

- The discussion of the results should be done with more rigor and consistency. Also should be longer.

For all the above comments I propose a major revision.

Reviewer #2: Comments to the authors

The proposed manuscript is very interesting and has a noteworthy objective. Congratulations to the authors.

My suggestions follow below.

Comment 1:

Abstract: In the phrase “Balance training is an unexplored component….”, do the authors mean that “Balance training has not been sufficiently explored…?” There are already some studies in this field.

Comment 2:

Keywords: I suggest not repeating the keyword that already exists in the title to enhance search results.

Comment 3:

The two phrases contained in lines 79-82 require references. The same goes for other sentences without references, specifically in the introduction section.

Comment 4:

The paragraph starting at line 79 and ending at 102, is very interesting for the study to point out the importance of balance in badminton, however the transition between the studies should be further explored so that its clearer for the reader.

Comment 5:

Line 103: “footwork performance” is core in your study, however, no refence in the introduction exists to introduce the importance of “footwork performance” neither how it is assessed. I consider its essential to add a paragraph to support “footwork performance” importance and assessment.

Comment 6:

Line 114: This is not an RCT. It may however be a clinical trial with a convenience sample which was randomized in to two groups.

Comment 7:

Was a sample size calculated? Please state this in the manuscript.

Comment 8:

Was data protection guaranteed? Please state this detail on the manuscript.

Comment 9:

Line 176: Please state the reference used “Each participant’s dominant leg was identified as the leg which moved close to the racket arm when lunging”.

Comment 10:

Line 216: please state the reference for the “Shuttle Run Test” used in the study.

Comment 11:

Line 225: Please state the reference for the “Measurement of push-off time in stroke play”.

Comment 12:

Statiscal analysis

Please state if parametric or non-parametric test were used in each data set analysis.

Comment 13:

Line 257: Please add P value calculation and if possible confidence intervals to “Table 1. Sample characteristics.”

Comment 14:

Line 266: It is not correct to refer a “trend” in the phrase “There was a trend for a time main effect (p = 267 0.114) and time x group interaction (p = 0.085) in shuttle run time:” It is simply not significant.

Comment 15:

Please state the limitations of the study in the manuscript.

Comment 16:

Conclusion may need reformulation as the sample size is considerably small and the fact that this study is not a RCT.

Comment 17:

Flow diagram seems incorrect in the Allocation section.

Comment 18:

Was the trial registered in a clinical trials platform?

6. PLOS authors have the option to publish the peer review history of their article (what does this mean?). If published, this will include your full peer review and any attached files.

Reviewer #1: No

Reviewer #2: No

---

## [Author Response · Author response to Decision Letter 0]

4 Oct 2022

Reviewer 1: Thank you very much for your constructive feedback on our manuscript. We have addressed all your suggestions and incorporated in to the revision. Thank you 

Reviewer 2: Thank you very much for your constructive feedback on our manuscript. We have addressed all your suggestions and incorporated in to the revision. Thank you

---

## [Decision Letter · Decision Letter 1]

3 Nov 2022

Effect of balance training on footwork performance in badminton: an interventional study

PONE-D-22-21470R1

Dear Dr. Malwanage,

We’re pleased to inform you that your manuscript has been judged scientifically suitable for publication and will be formally accepted for publication once it meets all outstanding technical requirements.

Kind regards,

Javier Abián-Vicén, Ph.D.

Academic Editor

PLOS ONE

Additional Editor Comments (optional):

Congratulations for your work!

Reviewers' comments:

Reviewer's Responses to Questions

**Comments to the Author**

1. If the authors have adequately addressed your comments raised in a previous round of review and you feel that this manuscript is now acceptable for publication, you may indicate that here to bypass the “Comments to the Author” section, enter your conflict of interest statement in the “Confidential to Editor” section, and submit your "Accept" recommendation.

Reviewer #1: All comments have been addressed

2. Is the manuscript technically sound, and do the data support the conclusions?

Reviewer #1: Yes

3. Has the statistical analysis been performed appropriately and rigorously? 

Reviewer #1: Yes

4. Have the authors made all data underlying the findings in their manuscript fully available?

Reviewer #1: Yes

5. Is the manuscript presented in an intelligible fashion and written in standard English?

Reviewer #1: Yes

6. Review Comments to the Author

Reviewer #1: Congratulations for your paper. I think the document meets the criteria established by Plos One to be published.

7. PLOS authors have the option to publish the peer review history of their article (what does this mean?). If published, this will include your full peer review and any attached files.

Reviewer #1: No

---

## [Editor Report · Acceptance letter]

7 Nov 2022

PONE-D-22-21470R1 

Effect of balance training on footwork performance in badminton: an interventional study 

Dear Dr. Malwanage:

I'm pleased to inform you that your manuscript has been deemed suitable for publication in PLOS ONE. Congratulations! Your manuscript is now with our production department. 

Kind regards, 

on behalf of

Dr. Javier Abián-Vicén 

Academic Editor

PLOS ONE